# Single-Cell RNA-Sequencing Reveals the Skeletal Cellular Dynamics in Bone Repair and Osteoporosis

**DOI:** 10.3390/ijms24129814

**Published:** 2023-06-06

**Authors:** Sixun Wu, Shinsuke Ohba, Yuki Matsushita

**Affiliations:** 1Department of Cell Biology, Nagasaki University Graduate School of Biomedical Sciences, Nagasaki 852-8588, Japan; bb55321202@ms.nagasaki-u.ac.jp (S.W.); ohba.shinsuke.dent@osaka-u.ac.jp (S.O.); 2Department of Tissue and Developmental Biology, Osaka University Graduate School of Dentistry, Osaka 565-0871, Japan

**Keywords:** aging, bone marrow adipocytes, bone marrow stromal cells (BMSCs), bone regeneration, lineage-tracing, osteoporosis, single-cell RNA-sequencing (scRNA-seq), skeletal stem and progenitor cells (SSPCs)

## Abstract

The bone is an important organ that performs various functions, and the bone marrow inside the skeleton is composed of a complex intermix of hematopoietic, vascular, and skeletal cells. Current single-cell RNA sequencing (scRNA-seq) technology has revealed heterogeneity and sketchy differential hierarchy of skeletal cells. Skeletal stem and progenitor cells (SSPCs) are located upstream of the hierarchy and differentiate into chondrocytes, osteoblasts, osteocytes, and bone marrow adipocytes. In the bone marrow, multiple types of bone marrow stromal cells (BMSCs), which have the potential of SSPCs, are spatiotemporally located in distinct areas, and SSPCs’ potential shift of BMSCs may occur with the advancement of age. These BMSCs contribute to bone regeneration and bone diseases, such as osteoporosis. In vivo lineage-tracing technologies show that various types of skeletal lineage cells concomitantly gather and contribute to bone regeneration. In contrast, these cells differentiate into adipocytes with aging, leading to senile osteoporosis. scRNA-seq analysis has revealed that alteration in the cell-type composition is a major cause of tissue aging. In this review, we discuss the cellular dynamics of skeletal cell populations in bone homeostasis, regeneration, and osteoporosis.

## 1. Introduction

Progressive aging has a substantial negative impact on the skeletal system. Diseases of the skeletal system associated with aging, particularly those caused by the loss of bone homeostasis and imbalances in bone metabolism, such as osteoporosis and related bone fractures, significantly shorten a healthy life span [1,2,3]. Cellular senescence is a key contributor to the imbalance in bone homeostasis [4]. Senescent cells accumulate in tissues with age, leading to an overall decline in tissue regeneration potential, which is causally linked to degenerative diseases [5,6]. Elimination of senescent cells prevents age-related bone loss in mice [7]. Therefore, dynamic changes in skeletal cells are strongly implicated in skeletal aging.

To address these bone-related negative events, a comprehensive understanding of skeletal cell biology is essential. Skeletal cells are composed of multiple cell types, including skeletal stem cells (SSCs), skeletal progenitor cells, differentiated chondrocytes, osteoblasts, and marrow adipocytes. SSCs are bone tissue-specific stem cells and are located at the pinnacle of the hierarchy of skeletogenesis. Over 50 years have passed since bone marrow SSCs were first established [8]. SSCs have been defined as mesenchymal cells, which have the potential of self-renewal and tri-lineage differentiation into chondrocytes, osteoblasts, and adipocytes [9,10,11]. They have been experimentally proven using ex vivo cell culture and/or subsequent in vivo ectopic transplantation. Although ex vivo primary cell culture experiments have been the gold standard for proving the property of SSCs, it is controversial whether these cell culture results alone can define cultured cells as stem cells. The target cells do not always behave as they would in the native biological environment since the culture conditions are quite different from those in the body. In addition, tri-lineage differentiation potential into chondrocytes, osteoblasts, and adipocytes, which is one of the criteria of SSCs, is evaluated under artificial conditions using distinct differentiation induction media. Hence, we should carefully define SSCs and their lineage cells by using several ways well-established in the research field of stem cell biology.

It has long been proposed that bone marrow contains multiple types of SSCs, their progenitor cells (SSPCs: skeletal stem and progenitor cells), and their lineage cells; only 0.001% to 0.01% of bone marrow cells contain stem cell populations [12]. Among the skeletal cells in bone marrow, bone marrow stromal cells (BMSCs) have been defined as mesenchymal populations, which are located between the outer surfaces of marrow blood vessels and bone surfaces [13], suggesting that multiple unclarified skeletal cell types are included in BMSCs. It has been reported that BMSCs are differentiated from SSPCs [14,15,16], while a subpopulation of BMSCs provides downstream skeletal cells as SSPCs [17,18]. These seemingly opposite roles of BMSCs are complicated. This is because BMSCs themselves have not been completely understood, including their spatial location. Consequently, distinct tissue-specific BMSCs need to be well-defined through basic research. Currently, BMSCs have been elucidated to be heterogeneous populations by using new technology [19], and will take us into a new era in skeletal stem cell biology.

Technological breakthrough has provided new insights into skeletal biology. Cell surface marker-based fluorescence-activated cell sorting (FACS) analysis and in vivo lineage-tracing studies are integral methods to investigate cell populations. These approaches enable us to define cellular populations and trace their progeny [20]. To adequately characterize a heterogeneous skeletal cell population, single-cell RNA-sequencing (scRNA-seq) analysis is a key technique to complement the previous approaches. scRNA-seq can reveal the molecular signature of individual cells, identify novel cell types, and provide insights into the cellular dynamics in tissues [21]. Advances in scRNA-seq technology in recent years have validated its reliability in identifying cell types and analyzing gene expression patterns [22].

In this review, based on scRNA-seq analysis and subsequent validation approaches, we discuss the role of distinct skeletal cell populations in bone regeneration and diseases, such as age-related osteoporosis.

## 2. Spatiotemporal-Specific Skeletal Stem Cells

SSCs play important roles in the development, maintenance, and regeneration of bone tissues. SSCs have the potential of self-renewal and differentiation into osteoblasts, chondrocytes, adipocytes, and stromal cells, which are necessary for bone growth and repair [9,10,11]. The concept of SSCs originated from studies in which autologous fragments of bone marrow or its cell suspensions were found to generate skeletal tissue after heterotopic transplantation [8,23]. Currently, SSCs in mouse long bones are identified by two distinct approaches: FACS-based isolation using appropriate cell surface markers, and an in vivo lineage-tracing approach. These approaches have the benefit of excluding hematopoietic and endothelial cells and can select target cells. In the former, mouse SSCs (mSSCs) are defined as those having several markers, including CD51^+^CD90^−^CD105^−^CD200^+^ [24], PDGFRα^+^Sca1^+^CD45^−^TER119^−^ [25], and CD73^+^CD31^−^ [26]. The latter, in vivo lineage-tracing approach using mouse genetic models, has revealed several types of SSCs at distinct locations in long bones, such as parathyroid hormone-related protein (PTHrP)^+^ growth plate stem cells residing within the resting zone of the growth plate in the early postnatal stage [27], cathepsin K^+^ periosteal stem cells in the early developmental stage [28], CXCL12^+^LepR^+^ reticular stromal cells in adult to aged bone marrows [17,18], and fibroblast growth factor receptor 3 (FGFR3)^+^ endosteal stromal cells in young bone marrows [29]. Notably, these populations targeted by distinct *cre* or *creER* lines are supposed to include the abovementioned SSC population. These *cre* driver lines not only mark SSCs but also other cell types. These lineage-tracing-based SSC identification processes are often performed together with the FACS-based cell surface marker isolation approach. This combined approach can identify bona fide SSCs along with their different spatial allocations and cellular dynamics [27,28,29]. These small populations of highly clonogenic SSCs present in each bone compartment play important roles in bone maintenance and regeneration.

## 3. Heterogeneity of BMSCs by scRNA-Seq

BMSCs are a versatile mesenchymal cell population supporting key skeletal functions. A part of the mesenchymal cells, residing in the perivascular regions surrounding sinusoids or distal arteries, are believed to contain SSPC populations [18,30,31] (Figure 1). In fact, human skeletal progenitor cells in the bone marrow have been reported to be located in the perisinusoidal area [9]. The majority of BMSCs are reticular in shape and adjacent to sinusoidal vessels. These reticular cells express C-X-C motif chemokine ligand 12 (CXCL12, also known as stromal cell-derived factor 1, SDF1) [32,33,34], leptin receptor (LepR) [17,34], stem cell factor (SCF, also known as KIT ligand) [35], and early B-cell factor 3 (Ebf3) [18]. Among all skeletal cells, these reticular stromal cells are one of the major populations in the bone marrow and are considered to possess various properties. Reticular stromal cells represent 0.3% of all bone marrow cells, including hematopoietic, vascular, and skeletal cells. CXCL12- and LepR-expressing reticular stromal cells predominantly occupy the marrow cavity among all skeletal cells [17,34]. These CXCL12^+^LepR^+^ reticular cells have been defined as total BMSCs. LepR^+^ cells in adult bone marrow almost entirely overlap (approximately 90%) with CXCL12^+^ cells [34,36]. They maintain a hematopoietic microenvironment by expressing cytokines and are in physical contact with each other; in addition, they are critical sources of hematopoietic stem cell niche factors [37,38,39]. These cells are required for the proliferation of hematopoietic stem cells, and lymphoid and erythroid progenitors [37]. In addition, they are a major source of bone cells and adipocytes in adult bone marrow and functionally regulate osteogenesis and adipogenesis [38,40,41,42,43]. In recent years, it has become clear that BMSCs are heterogeneous, thereby making it difficult to accurately describe them by the umbrella term, BMSCs.

scRNA-seq is a progressive technology used to characterize heterogeneous cell populations, allowing rapid mapping of the BMSC landscape. scRNA-seq studies have identified several types of BMSCs. Baryawno et al. identified mesenchymal stromal cells (showing high expression of LepR), stroma-descendent osteolineage cells, bone marrow-derived endothelial cells, and other stromal cell types [44]. Tikhonova et al. performed scRNA-seq analyses of the bone marrow microenvironment at a steady state and identified two vascular endothelial-cadherin (VE-Cad)^+^ endothelial cells, four LepR^+^ perivascular, and three 2.3 kb type 1 collagen (COL2.3)^+^ osteolineage clusters, and a subpopulation of proliferative cells [19]. Analyses of the formation of fibroblastic colony-forming units (CFU-F) and the trajectory of differentiation revealed that the LepR^+^ compartment possesses multilineage differentiation potential and SSC activity.

To accurately map BMSCs, Zhong et al. performed scRNA-seq analyses using *Col2a1-cre* mouse long bones at 1, 3, and 16 months. They found subpopulations of BMSCs, including clusters of early, intermediate, and late mesenchymal progenitors and lineage-committed progenitors [40]. Interestingly, the pseudotime trajectory represented early mesenchymal progenitors expressing SSC marker genes at one end, whereas osteoblasts and adipocytes are at the opposite and distinct ends, thereby implying that a subset of BMSCs has the potential to bi-differentiate into osteogenic and adipogenic lineages in physiological conditions. In addition, LepR^+^ cells in the adult bone marrow consist of mesenchymal progenitors and a unique cell type that expresses adipocyte markers, including *Adipoq* but does not contain lipid droplets. The adipogenic cell type is termed the bone marrow adipogenic lineage precursor (MALP). Wolock et al. performed scRNA-seq to analyze non-hematopoietic (CD45^−^/Ter119^−^) and non-epithelial/non-endothelial cells (CD31^−^) and identified that CXCL12-abundant reticular (CAR) cells are highly enriched in multipotent stromal cells, adipocyte-progenitor cells, and osteoblast progenitors [45]. scRNA-seq analysis of CXCL12^+^ BMSCs using young *Cxcl12-GFP* mice identified two major populations, including preadipocyte-like reticular cells: Adipo-CAR cells and preosteoblast-like cells named Osteo-CAR cells [34]. Interestingly, these two populations have been predicted to be quite distinct [38]. *Lepr-cre* labels most BMSCs and osteogenic lineage cells in adult long bones, and these BMSCs include adipogenic populations, such as Adipo-CAR cells or MALPs as per scRNA-seq analysis of young adult and aging *Lepr-cre* mice [46].

Paired-related homeobox protein 1 (Prrx1) is a transcription factor that is prominently expressed in the mesenchyme during the crucial stages of craniofacial and limb development [47]. Thus, *Prrx1-cre* labels all limb skeletal cells in the appendicular skeleton [48]. Currently, scRNA-seq for *Prrx1-cre* lines at young and aged stages revealed the transitional stromal populations between typical skeletal cell types (Figure 1). Through the process of single-cell data downscaling and visualization analysis by using UMAP plots, osteoblast–chondrocyte transitional (OCT) cells and osteoblast–reticular transitional (ORT) cells, which have intermediate properties of typical gene expressions of both, are identified as a type of BMSCs. Among them, *Fgfr3* is highly expressed in the OCT cluster. ORT cells marked by growth arrest-specific 1 (*Gas1*), regarded as a cell cycle inhibitor [49], are located between osteoblast and *Cxcl12* highly expressed reticular populations. Bioinformatics investigation using RNA velocity analysis revealed that the OCT population provides osteoblasts, ORT cells, and pre-adipocyte-like reticular stromal cells at the young stage. In contrast, these once differentiated reticular stromal cells behave as the origin of skeletal cells at an aged stage [29], suggesting that cellular origins’ shift of BMSCs may occur with the advance of age. This idea could be the key to uncovering the biology of BMSCs (Figure 1).

Taken together, these scRNA-seq approaches elucidate the heterogeneity of BMSCs. They are grouped into several specific types, such as osteogenic and chondrogenic bi-potential mesenchymal progenitors, osteogenic and adipogenic progenitors, and osteo- or adipose-lineage-determined progenitors. Importantly, spatiotemporal multi-potential SSPCs are included in a subset of BMSCs. However, it is worth noting that scRNA-seq analysis can only provide an approximate representation of the diverse cell populations within the BMSCs. It cannot strictly show the interactions with peripheral vascular and hematopoietic cells. In addition, currently published scRNA-seq analyses have lost spatial information. Therefore, it is imperative to develop a strategy for high-dimensional integrated analyses and spatial transcriptomic analyses, which approach will enable a more comprehensive understanding of the complex cellular interactions and dynamics within the bone marrow stromal microenvironment. Furthermore, we strongly believe that enhancing the accuracy of single-cell sequencing analysis can be achieved by effectively integrating the biological information acquired from in vivo cell profiling with single-cell sequencing analysis. This synergy holds great potential for improving our understanding of the complex cellular landscape and enhancing the reliability of single-cell sequencing data.

## 4. Cellular Dynamics of BMSCs

Bone is a dynamic and non-stop tissue that undergoes constant remodeling throughout our lives [50]. Multiple types of cells play a crucial role in this process by regulating bone remodeling similar to osteocytes [51]. Consistent with these scRNA-seq results, biological data using an in vivo lineage-tracing study with tamoxifen-inducible *creER* or constitutively active *cre* lines show the heterogeneity and the dynamics of BMSCs. Multiple studies address the dynamics of a major BMSCs population, which expresses CXCL12 and LepR [17,44]. *Lepr-cre*-marked reticular stromal cells are the major source of osteoblasts and adipocytes in adult bone marrow. These cells are in the bone marrow and are hardly detected in bones at 2 months of age. However, Lepr-cre^+^ osteoblasts gradually increase from 6 to 14 months of age [17]. A lineage-tracing study using a tamoxifen-inducible *Lepr-creER* line revealed that Lepr-creER^+^ marrow stromal cells at the perinatal stage decrease and do not differentiate into osteoblasts with the advance of age. In contrast, adult Lepr-creER^+^ cells become the main source of osteoblasts [43]. A transcription factor Ebf3 is expressed in CXCL12^+^LepR^+^ BMSCs and Ebf3^+^ cells marked by *Ebf3-creER* behaving as SSPCs in adult bone marrow [18]. Hence, CXCL12^+^LepR^+^ reticular stromal cells play a role as SSPCs in the adult stage. The *Cxcl12*-creER line has revealed the unique cell fate of an adipogenic subset of CXCL12^+^LepR^+^ reticular stromal cells called Adipo-CAR cells. Cxcl12-creER precisely marks a relatively quiescent subset of CXCL12^+^LepR^+^ cells in the central marrow space after a tamoxifen injection [34]. Adipo-CAR cells spontaneously differentiate into Perillipin^+^ marrow adipocytes. Other adipogenic reticular stromal cells, called MALPs, also contribute to marrow adipocytes. Short-term ablation of CXCL12^+^LepR^+^ cells in vivo using diphtheria toxin significantly reduces the number of hematopoietic stem and progenitor cells [37]. BMSC ablation using *Adipoq*-cre; DTR mouse with diphtheria toxin injection showed de novo trabecular formation [40]. Interestingly, LepR deletion in BMSCs using *Prrx1*-cre increases osteogenesis and decreases adipogenesis [41], whereas CXCL12 deletion using *Prrx1*-cre and *Osx*-cre decreases osteogenesis and increases adipogenesis [42]. A study of *Adipoq-creER* mice revealed that Perillipin^−^ MALPs differentiate into Perillipin^+^ adipocytes in the marrow space [40]. These Cxcl12-creER^+^ cells are dormant and remain in primitive regions of the bone marrow space, suggesting that adipogenic Cxcl12-creER^+^ cells may be derived from other mesenchymal precursor cells, such as type II collagen^+^ cells adjacent to the growth plate or PTHrP^+^ resting chondrocytes [27,52] as scRNA-seq data mentions the existence of their upstream cells [40]. In fact, CXCL12^+^LepR^+^ BMSCs have been proposed to be differentiated cells, which are located downstream of SSPCs during development [14,15,27,52,53,54]. Several studies have elucidated the origins of these reticular cells. Early postnatal mesenchymal cells in the metaphyseal area marked by *Gli1-creER*, *Pdgfrb-creER*, and *Lepr-creER* contribute to reticular cells [15,55]. Early postnatal Osx^+^ cells also transform into whole reticular cells [14]. Moreover, in the embryonic developmental stage, chondrogenic cells inside the cartilage template marked by *Fgfr3-creER* become postnatal metaphyseal reticular cells [16]. In contrast, the outer layer of perichondrial cells marked by *Dlx5-creER* surrounding the cartilage template contributes to diaphyseal reticular cells, although the Osx^+^ inner layer of perichondrial cells contribute to skeletal cells transiently [16]. Recently, endosteal stromal populations were spatiotemporally identified. Juvenile Fgfr3-creER^+^ cells located in the endosteal surface robustly contribute to CXCL12^+^LepR^+^ reticular stromal cells and osteoblasts.

In summary, BMSC heterogeneity has been elucidated. The central parts of BMSCs are marked by CXCL12 and LepR. They include multipotent progenitors, pre-osteogenic BMSCs (Osteo-CAR cells), and pre-adipogenic BMSCs (Adipo-CAR cells and MALPs). These BMSCs behave as SSPCs in adult and aged stages. In contrast, Fgfr3^+^ endosteal BMSCs contribute to skeletogenesis mainly at the young stage.

## 5. Role of BMSCs on Bone Regeneration

The in vivo lineage-tracing approach is a useful tool to visualize cellular dynamics during bone regeneration. Many types of skeletal cell-specific *cre* or *creER* mice have been used to elucidate the contribution of SSPCs to the bone regeneration process. However, many of these models include “constitutively active” cre, such as *Prrx1-cre* [56] and *Ctsk-cre* [28] for periosteum, *Lepr-cre* [17], *Adipoq-cre* [57], and *Mx1-cre* [58] for bone marrow. In bone regeneration processes, it is important to strictly control the timing of cre recombination because various genes are positively and negatively expressed during the healing phase and there are dynamic changes in the cellular state [59]. The use of constitutively active cre may not accurately capture the dynamics of the cells labeled prior to bone regeneration because the expression of the driver gene for recombinase may be upregulated during the bone regeneration process, possibly leading to new recombination during bone regeneration. However, tamoxifen-inducible CreER lines are not affected by drastic changes in gene expression at the site of bone regeneration because the timing of recombination can be controlled. Therefore, the lineage of skeletal cells originally present in the tissue can be accurately investigated using CreER systems.

CXCL12^+^LepR^+^ BMSCs marked by *Lepr-creER* in adult mice contribute to regenerative osteoblasts and chondrocytes upon bone fracture [43]. *Cxcl12-creER* marked Adipo-CAR cells, a subset of CXCL12^+^LepR^+^ cells, robustly differentiate into regenerative osteoblasts during the bone regeneration process, unlike that observed in physiological conditions. Interestingly, scRNA-seq analyses of these regenerating cells have revealed that differentiated Adipo-CAR cells de-differentiate into SSC-like cells in response to injury and re-differentiate into osteoblasts, thereby contributing to bone regeneration. This demonstrates that Adipo-CAR cells are a promising source for bone regeneration. Simultaneously, in addition to Adipo-CAR cells, various populations, including osteoblasts and their progenitors, participate in bone regeneration. Dlx5-creER^+^ osteoblast progenitor cells contribute to the regenerating bone. Osx-creER^+^ osteoblasts also contribute similarly, albeit in small amounts. It has been considered that MSCs or SSCs, which are at the top of the differentiation hierarchy, are responsible for bone regeneration by being the source of all differentiated cells. However, current studies reveal that a new mechanism exists wherein various cells, such as BMSCs and osteoblast progenitor cells acquire SSC-like properties through plasticity and differentiate into osteoblasts, thereby causing bone regeneration [34].

Various signaling pathways are involved in bone regeneration. Canonical Wnt signaling is one of the critical pathways for the process. Bone healing after an injury is accelerated in global Axin2^LacZ/LacZ^ mice, wherein the cellular response to Wnt is increased [60], such that Wnt signaling promotes bone regeneration. Since several types of cells are involved in bone regeneration, as revealed by cellular plasticity, cell type-specific analyses are needed to clarify the precise mechanism underlying the action of Wnt signaling. Wnt signaling activated by a transcriptional switch in cells of the skeletal lineage in vivo is highly dependent on the differentiation stage [61]. Single-cell RNA-seq analyses and lineage-tracing experiments reveal that Cxcl12-creER^+^ BMSCs transform into osteoblast precursor cells in a process mediated by canonical Wnt signaling. Similarly, Dlx5-creER^+^ osteoblast progenitor cells are mediated by Wnt signaling. In general, SRY-box transcription factor 9 (Sox9) and runt-related transcription factor 2 (Runx2) are master transcription factors for skeletal development. However, Cxcl12-creER^+^ BMSCs transform into osteoblast precursor cells in a process mediated by canonical Wnt signaling instead of Sox9-related or Runx2-related pathways [34].

The CXCL12/CXCR4 chemokine ligand/receptor axis plays an essential role in bone fracture healing. Deletion of CXCR4 in Tie2-cre^+^ endothelial progenitor cells delays the healing process, while CXCL12-injected WT mice heal significantly faster with sufficient callus formation than non-injected control mice [62]. Bone morphogenetic protein 2 (BMP2) is required for the initiation of bone regeneration [63]. BMP signaling is essential for the formation of osteo-chondroprogenitor cells in the initial stage [64]. CXCL12^+^BMP2^+^ perivascular cells are recruited along the endosteum; thereafter, a timely increase in BMP2 leads to the downregulation of CXCL12, which is essential in determining the fate of the CXCL12^+^BMP2^+^ cells for osteogenesis while diverging from their supportive role for angiogenesis [65]. Furthermore, Notch signaling is necessary for bone regeneration; the deletion of the Notch signaling molecule, Rbpjk, using *Prrx1-cre* mice results in fracture non-union [66] These ideas are of great significance for the further improvement of stem cell-based regenerative medicine.

## 6. Characteristics of Aging BMSCs

Cellular identity, rather than the tissue environment, determines the degree of aging, suggesting that alterations in the composition of cellular identity are a major cause of tissue aging [67] Current scRNA-seq analyses have revealed single-cell characteristics and trajectories during the aging process. scRNA-seq studies on the skeletal system have shown that the aging of SSCs diminishes their transcriptomic diversity. Aging of SSCs changes the bone marrow niche environment and differentiation process of the skeletal lineage, thereby leading to the reduction of the potential of bone and cartilage formation. Interestingly, aged SSCs with youthful circulation, including young hematopoietic stem cells, do not show osteochondrogenic activity. In contrast, the aged SSC lineage influences hematopoietic aging [68].

Similarly, BMSCs undergo functional changes with increasing age. Fgfr3-creER^+^ endosteal stromal cells have a high colony-forming ability in young, but not in adult and aged stages. Instead, Lepr-cre^+^ central reticular stromal cells and their lineages increase the rate of colony-forming cells with the advance of age [29]. Functionally, the absence of Ebf3 in the aging BMSCs and their lineage cells marked by *Lepr-cre* causes osteosclerotic marrow cavity, indicating that Ebf3 maintains hematopoietic niches and the marrow cavity [18].

Although the BMSCs are multipotent like SSCs in the aged stage, they can only differentiate into osteoblasts and adipocytes in physiological conditions. As previously mentioned, CXCL12^+^LepR^+^ BMSCs produce osteogenic and adipogenic lineage cells, but their commitment to these two lineages is mutually exclusive [69]. Aging induces CXCL12^+^LepR^+^ BMSCs to differentiate into adipocytes rather than osteoblasts [69,70], which can explain the enhanced accumulation of bone marrow adipose tissue (BMAT) during aging [71]. The increase in BMAT under aging conditions may be attributed to the capacity of adipogenic BMSCs to easily convert to lipid-laden bone marrow adipocytes.

Several transcription factors and signaling pathways underlie the potential mechanisms of bone–fat imbalance and mesenchymal stromal cell exhaustion during skeletal aging (Figure 2). For example, peroxisome proliferator-activated receptor-γ (PPARγ) and CCAAT/enhancer-binding protein-α (CEBPα) contribute to the adipogenic differentiation of SSCs. Their expression is maintained at a high level throughout the differentiation process and during the entire lifespan of adipocytes [72,73]. Runx2 promotes osteogenesis and inhibits differentiation into adipocytes [74]. Parathyroid hormone (PTH) coordinates the signaling of local factors, including transforming growth factor beta (TGF-β), Wnt, and BMP, to regulate the fate and differentiation of SSCs [75]. CXCL12-deficient mesenchymal progenitor cells tend to contribute to bone marrow adipocytes [38], suggesting that CXCL12 suppresses the differentiation of mesenchymal progenitors into the adipogenic lineage.

Epigenetic factors also play key roles in the aging and the fate of stem cells. Lysine demethylase 4B (KDM4B) is an important epigenetic factor, which controls the self-renewal of stem cells and maintains a BMSC pool in vivo [69]. A loss of KDM4B in BMSCs leads to a reduction in bone formation and an increase in bone marrow adiposity [69]. However, further studies are required to confirm the identity of these BMSCs and the mechanisms associated with the aging of bone marrow.

Aging also has a profound effect on bone repair. Bone repair capacity is significantly reduced upon aging, and macrophages from aged mice show increased levels of M1/inflammatory genes and dysregulation of other immune-related genes during fracture repair [76]. In addition, studies using *Actin-creER* have revealed that clonal activity is reduced upon aging during the bone regeneration process. Aged mice have a reduced SSC population (CD51^+^CD200^+^CD90^−^CD105^−^) in this process [68].

## 7. Effect of BMSC Aging on Osteoporosis

Bone strength is reduced by osteoporosis during aging and is also severely affected by congenital metabolic diseases, such as osteogenesis imperfecta and osteomalacia [3]. Osteoporosis, caused by diseases and medications, is divided into primary osteoporosis, including juvenile, postmenopausal, and senile and secondary osteoporosis [77]. Primary osteoporosis is more common than secondary osteoporosis. Senile osteoporosis is an age-related disease caused by an imbalance between bone formation and resorption. In physiological conditions, bone formation and resorption take place to an extent comparable to each other, which is known as bone remodeling. A decrease in bone formation and/or an increase in bone resorption are biological features of osteoporosis. Reduced bone formation primarily owing to decreased function of osteoblast and increased BMAT has become a notable health problem worldwide [78,79]. Alterations in the number and function of BMSCs are an important cause of senile osteoporosis [80,81,82]. The proportion of senescent BMSCs in the skeletal microenvironment increases with age, thereby leading to a decrease in the osteogenic capacity of osteoblasts [81,83]. Notably, only a small fraction of cells undergo senescence even at advanced ages. For example, approximately only 11% of osteocytes in mouse bones undergo senescence with age [84]. Although the number of senescent cells is relatively low, tissue damage is profound. Furthermore, senescent cells from the osteoblast lineage produce a unique senescence-associated secretory phenotype (SASP) signal that leads to the over-secretion of a complex array of cytokines, chemokines, and matrix-degrading proteases, thereby creating a toxic microenvironment that results in age-related bone loss [85,86]. In contrast, BMSCs normally differentiate into osteoblasts and adipocytes; however, aging BMSCs differentiate into osteoblasts to a significantly lesser extent than adipocytes, presenting a negative correlation between bone mass and marrow adiposity, which is one of the main causes of senile osteoporosis [69] (Figure 2). In addition, bone marrow adipocytes can exert paracrine inhibitory effects on the osteogenic differentiation of BMSCs by blocking BMP signaling, and this mechanism may be mediated by adipokine-induced activation of the nuclear factor kappa beta (NF-κB) pathway [87]. Therefore, both SASP production and the abnormal differentiation of senescent BMSCs lead to a decrease in the osteoblast population, resulting in reduced bone formation and age-related osteoporosis. Delayed maturation of osteoblasts is also an integral cytological basis for the development of osteoporosis [81,88,89]. Most studies support that BMSCs differentiate into adipocytes with aging by increasing PPARγ expression, thereby resulting in decreased bone formation and osteoporosis [90,91]. In contrast, PPARγ conditional deletion in all skeletal cells using a *Prrx1-cre* mouse genetic model revealed that PPARγ increases bone marrow adiposity but is dispensable for the loss of cortical and trabecular bone tissue caused by aging [91,92]. A single-cell landscape of osteoporosis showed that the majority (76.3%) of the osteoblast lineages were blocked at the premature osteogenic stage and these delayed premature osteoblasts exhibited activation of cell senescence [93].

## 8. Conclusions

In this review, we discussed the heterogeneity of BMSC lineages based on scRNA-seq data, with a particular focus on the unique behavior of distinct subpopulations of BMSCs under physiological, injury, and aging conditions and the heterogeneity associated with age-related osteoporosis.

Adipogenic BMSCs, a subpopulation of central reticular stromal cells, slowly differentiate into bone marrow adipocytes during aging. These cells remain within the bone marrow space in a dormant state and do not migrate or differentiate into cortical bone osteoblasts [34]. Upon injury, these cells can be recruited to the cortical defect and differentiate into osteoblasts during cortical bone repair in a canonical Wnt signaling pathway-dependent manner [94]. Senescent cells secrete proinflammatory factors, and the formation of an abnormal bone marrow microenvironment aggravates bone loss. A decrease in osteoblast count and a significant increase in bone marrow adipocyte count are the characteristic pathological features of osteoporosis [69].

With the application of single-cell sequencing technology, the heterogeneity and trajectory of BMSCs have become clear [19,44,45]. LepR^+^ BMSCs with adipogenic potential consist of at least two types of adipose and bone precursor cell populations that can differentiate into osteogenic and adipogenic cells [34]. Bone marrow adipocytes and osteoblasts are derived from common mesenchymal progenitors but are committed to their distinct lineages in a mutually exclusive manner [71,95]. A study of transcriptional and epigenomic changes in the adipocyte and osteoblast lineages of human mesenchymal stem cells revealed that osteogenesis primarily involves the activation of preestablished enhancers, whereas adipogenesis involves de novo establishment and activation of enhancers [96].

Intrinsic cellular senescence leads to the loss of transcriptome diversity in BMSCs, ultimately distorting the output of skeletal and hematopoietic lineages, diminishing skeletal integrity, and limiting healing capacity. Senescent BMSCs have been proposed to tend to differentiate into bone marrow adipocytes during aging, and then, contribute to osteoporosis [69]. Aging mesenchymal progenitors exhibit DNA damage, and their PPARγ expression is significantly increased, and then marrow adipogenesis is induced [97]. However, several studies suggested that increased marrow adipogenesis does not contribute to bone loss [92]. Therefore, further studies are needed to investigate the relationship between BMSC adipogenesis and osteoporosis.

Upon bone marrow injury, BMSCs differentiate into osteoblasts and promote bone regeneration. Revealing the cellular dynamics of the response of senescent BMSCs to injury or the descendant cells from the BMSCs after injury is a challenge. The clearance of senescent cells delays aging-associated disorders [98]. Therefore, cell therapy presents a feasible solution to treat age-related diseases such as senile osteoporosis. Osteoporosis increases the risk of bone fractures, which often leads to bedridden conditions in patients. Understanding the mechanism of fractures in aged individuals is important as the world population is aging, and the number of patients with osteoporosis is expected to increase in the future.

## Figures and Tables

**Figure 1 ijms-24-09814-f001:**
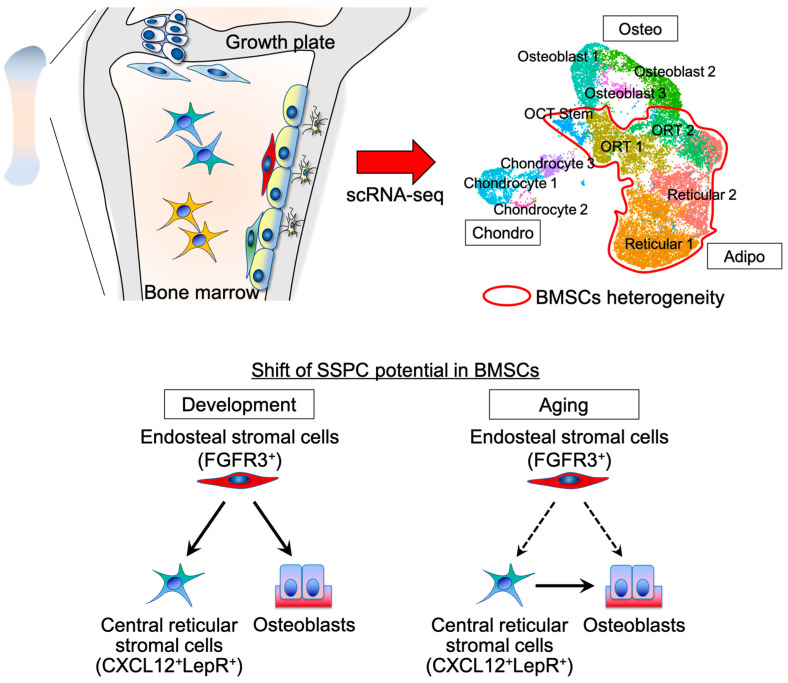
BMSC heterogeneity and the shift of SSPC potential. scRNA-seq analysis reveals the heterogeneity of BMSCs. Fgfr3^+^ endosteal cells behave as SSPCs in the growing stage. They differentiate into osteoblasts and CXCL12^+^LepR^+^ reticular stromal cells. In the aging stage, CXCL12^+^LepR^+^ reticular stromal cells contribute to osteoblasts.

**Figure 2 ijms-24-09814-f002:**
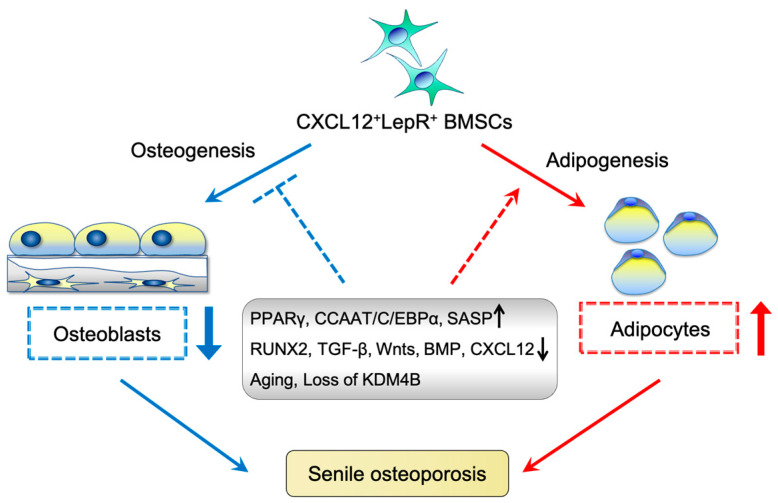
A series of drivers influence the lineage of CXCL12^+^LepR^+^ BMSCs, causing bone–fat imbalance and leading to senile osteoporosis. Several transcription factors, epigenetic factors, signaling pathways, and physiological phenomena promote adipogenic differentiation while inhibiting osteogenic differentiation. They reduce bone formation and increase bone marrow adiposity, ultimately contributing to senile osteoporosis.

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
