# Peer review of "Single-Cell RNA-Sequencing Reveals the Skeletal Cellular Dynamics in Bone Repair and Osteoporosis"

_ijms, 2023, doi:10.3390/ijms24129814_

Round 1

Reviewer 1 Report

This is a very well-designed and presented review. I appreciate the author's work in putting the evidence together in this manuscript. I only have some minor comments and recommendations.

1) Figure 1 and 2 are not referenced in the main text

2) The authors can consider addressing gaps and limitations in the research on BMSCs. For example, they can discuss the challenges and limitations in studying BMSCs using Single-cell RNA-sequencing

3) The authors can consider providing more specific and detailed recommendations for future research. For instance, they can suggest specific experimental designs, techniques, or approaches that can help address the remaining questions and challenges in the field.

Author Response

Comment 1:

              -Figure 1 and 2 are not referenced in the main text.

Author Response:

              Thank you for your valuable comment. We have incorporated your suggestions and made the necessary revisions to the manuscript. Specifically, we have re-labeled the references previously referred to as Figure 1 and Figure 2. In line 90, we have removed the reference to Figure 1, and in lines 178 and 190, we have added the corresponding reference. Figure 2 is referenced in lines 347 and 404. Throughout the article, we have maintained consistent references to Figure 1 and Figure 2.

Comment 2:

              - The authors can consider addressing gaps and limitations in the research on BMSCs. For example, they can discuss the challenges and limitations in studying BMSCs using Single-cell RNA-sequencing.

Author Response:

              Thank you very much for the suggestion. In the revised manuscript, we added the challenges and limitations in studying BMSCs using Single-cell RNA-sequencing.

The following sentences were added (Line 195-199):

              However, it is worth noting that scRNA-seq analysis can only provide an approximate representation of the diverse cell populations within the BMSCs. It cannot strictly show the interactions with peripheral vascular and hematopoietic cells. In addition, currently published scRNA-seq analyses have lost spatial information.

Comment 3:

              - The authors can consider providing more specific and detailed recommendations for future research. For instance, they can suggest specific experimental designs, techniques, or approaches that can help address the remaining questions and challenges in the field.

Author Response:

              Thank you very much for the suggestion. We have now included the following sentences to propose specific experimental approaches to help solve the remaining problems and challenges in the field.

The following sentences were added (Line 199-207):

              Therefore, it is imperative to develop a strategy for high-dimensional integrated analyses and spatial transcriptomic analyses, which approach will enable a more comprehensive understanding of the complex cellular interactions and dynamics within the bone marrow stromal microenvironment. Furthermore, we strongly believe that enhancing the accu-racy of single-cell sequencing analysis can be achieved by effectively integrating the biological information acquired from in vivo cell profiling with single-cell sequencing analysis. This synergy holds great potential for improving our understanding of the complex cellular landscape and enhancing the reliability of single-cell sequencing data.

Reviewer 2 Report

Although a lot of research has been carried out to reveal dynamics of bone repair and osteoporosis, the current paper well discusses new aspects and the role of single-cell RNA-sequencing analysis in cell-type composition alteration and tissue aging.

Author Response

Comment:

-Although a lot of research has been carried out to reveal dynamics of bone repair and osteoporosis, the current paper well discusses new aspects and the role of single-cell RNA-sequencing analysis in cell-type composition alteration and tissue aging.

Author Response:

              Thank you for your valuable feedback. We appreciate your input, and we are committed to further enhancing the original draft based on your suggestions. Your comments have been immensely helpful, and we will work diligently to incorporate the necessary improvements to make the draft even stronger.

Reviewer 3 Report

The aim of this current review is to discuss the cellular dynamics of skeletal cell populations in bone homeostasis, regeneration, and osteoporosis. This research is under the scope of this journal; the topic is relevant for readers, and this research deals with potentially significant knowledge of the field. And It will be important for Biomedicine knowledge. The topic is relevant for readers and this study deals with potentially substantial knowledge of the field and opens new ways for future studies. 

However, there are some aspects which is needed to be improved in the manuscript:

(Keywords)

  • Please add keywords, and order the keywords / Mesh terms alphabetically
  • Bone remodelling represents the most remarkable bone response to mechanical stress and mineral homeostasis. It is the consequence of complex highly orchestrated and tightly regulated cellular processes taking place in a specialized entity - the Bone Remodelling Compartment (BRC). Please read the Brochado Martins, 2020 (Folia Morphologica Journal, 10.5603/FM.a2020.0134), in Remodelling compartment in root cementum (RCRC), Hypothesizing that similar cellular mechanisms underlie bone and cementum remodelling, the present work shows, for the first time, the histological evidence of a specialized remodelling compartment in dental hard tissues (Different from dentine). 
  • Bone is a non-stop tissue, please read also this article https://doi.org/10.3390/molecules26051339 to support the importance on dynamic tissue. 

Author Response

Comment 1:

              - Please add keywords, and order the keywords / Mesh terms alphabetically.

Author Response:

              Thank you very much for your suggestion. We have made further updates to the manuscript by including the relevant keywords and reordering them accordingly.

The following keywords were added (Line 24-26):

              Aging; bone marrow adipocytes; bone marrow stromal cells (BMSCs); bone regeneration; lineage-tracing; osteoporosis; single-cell RNA-sequencing (scRNA-seq); skeletal stem and progenitor cells (SSPCs)

Comment 2:

              - Bone remodelling represents the most remarkable bone response to mechanical stress and mineral homeostasis. It is the consequence of complex highly orchestrated and tightly regulated cellular processes taking place in a specialized entity - the Bone Remodelling Compartment (BRC). Please read the Brochado Martins, 2020 (Folia Morphologica Journal, 10.5603/FM.a2020.0134), in Remodelling compartment in root cementum (RCRC), Hypothesizing that similar cellular mechanisms underlie bone and cementum remodelling, the present work shows, for the first time, the histological evidence of a specialized remodelling compartment in dental hard tissues (Different from dentine).

Author Response:

              Thank you very much for your precious suggestion. We have now included the following sentences to discuss bone remodelling.

The following sentences were added (Line212-213):

              Multiple types of cells play a crucial role in this process by regulating bone remodeling similar to osteocytes.

Comment 3:

              -Bone is a non-stop tissue, please read also this article https://doi.org/10.3390/molecules26051339 to support the importance on dynamic tissue.

Author Response:

              Thank you very much for the suggestion. This study contributes valuable insights into the osteoconductive properties of collagenated porcine-derived bone graft materials and emphasizes the potential for developing different formulations to enhance bone healing. We have now included the following sentences in our manuscript.

The following sentences were added (Line210-211):

              Bone is a dynamic and non-stop tissue that undergoes constant remodeling throughout our lives

Reviewer 4 Report

In the manuscript titled “Single-cell RNA-sequencing reveals the skeletal cellular dynamics in bone repair and osteoporosis” the authors reviewed the role of distinct skeletal cell populations in bone regeneration and diseases based on scRNA-seq analysis. This review manuscript is written well and informative to the readers and the authors are appropriate for reviewing it because they published many important papers in this field. After some minor revision concerning the references, it will be acceptable for the publication.

Given that this is a review article, it would be advisable for the authors to limit the number of citations of other review articles.

Line 42-44, The authors need to replace the reference # 9, 10 which are review articles to original articles to show the definition of mesenchymal cells.

Line 56-60, Change the reference # 12, review article to an appropriate original article to show the location and the definition of BMSCs.

Line 83-85, The reference # 22-24 should be replaced to some appropriate original articles to describe the functions of SSCs.

Line 98, Please spell out the “FGFR3”.

Line 102-105, The authors need to refer some appropriate articles to these descriptions.

Line 167 and 172, To assist readers who are not familiar with this field, it would be helpful to spell out and provide a detailed explanation of Prrx1 and Gas1, including their definitions and a brief overview of their significance.

Line 171, A brief explanation on UMAP plots will be helpful for readers who are unfamiliar with scRNAseq.

Line 224-227, The authors should refer some articles to this description.

Line 266, The reference # 57 is inappropriate because this is not an original article but a “Hypothesis” article. Please delete or replace it to proper original articles.

Line 272-274, The authors need to change the reference # 59, 60, to some appropriate original articles.

Line 316, The reference # 67 should be replaced because this is a “perspective” article.

In the section of “Characteristics of aging BMSCs”, the authors need to re-consider the reference # 70, 71, 74, and 76 since they are review articles.

In the section of “Effect of BMSD aging on osteoporosis”, the reference # 79, 81, 82, 87, 88, 89, 91, and 92 should be replaced to proper original articles.

Line 391, The authors need to refer an appropriate article to the description of “these delayed premature osteoblasts exhibited activation of cell senescence.”

Author Response

Comment 1:

              -Given that this is a review article, it would be advisable for the authors to limit the number of citations of other review articles.

Author Response:

              Thank you very much for your precious suggestion. We have made the necessary revisions to the manuscript and effectively reduced the number of citations in the same type of review.

Comment 2:

-Line 42-44, The authors need to replace the reference # 9, 10 which are review articles to original articles to show the definition of mesenchymal cells.

Author Response:

              Thank you very much for your suggestion. We have referenced the original article.

The following references were added (Line 43-45):

              Sacchetti et al. 2007, Méndez-Ferrer et al. 2010, and Friedenstein et al. 1987.

Comment 3:

-Line 56-60, Change the reference # 12, review article to an appropriate original article to show the location and the definition of BMSCs.

Author Response:

              Thank you very much. We have changed the review article to the original article.

The following reference were added (Line 57-61):

              Bianco P and Boyde A. 1993

Comment 4:

-Line 83-85, The reference # 22-24 should be replaced to some appropriate original articles to describe the functions of SSCs.

Author Response:

              Thank you very much for your suggestion. We have changed the review articles.

The following references were added (Line 83-86):

              Sacchetti et al. 2007, Méndez-Ferrer et al. 2010, and Friedenstein et al. 1987.

Comment 5:

-Line 98, Please spell out the “FGFR3”.

Author Response:

              Thank you very much for your suggestion. We have made a revision to the manuscript.

The following words were added (Line 98-99):

              fibroblast growth factor receptor 3 (FGFR3)

Comment 6:

-Line 102-105, The authors need to refer some appropriate articles to these descriptions.

Author Response:

              Thank you very much for your suggestion. We have added some references to the manuscript.

The following references were added (Line 103-105):

              Mizuhashi et al., 2018; Debnath et al., 2018; Matsushita et al., 2023

Comment 7:

-Line 167 and 172, To assist readers who are not familiar with this field, it would be helpful to spell out and provide a detailed explanation of Prrx1 and Gas1, including their definitions and a brief overview of their significance.

Author Response:

              Thank you very much. We have now included the following sentences to help understand.

The following sentences were added (Line 174-177):

              Paired-related homeobox protein 1 (Prrx1) is a transcription factor that is prominently expressed in the mesenchyme during the crucial stages of craniofacial and limb development. Thus, Prrx1-cre labels all limb skeletal cells in the appendicular skeleton.

The following sentences were added (Line 182-184):

              ORT cells marked by Growth arrest specific 1 (Gas1), regarded as a cell cycle inhibitor, are located between osteoblast and Cxcl12 highly-expressed reticular populations.

Comment 8:

-Line 171, A brief explanation on UMAP plots will be helpful for readers who are unfamiliar with scRNAseq.

Author Response:

              Thank you very much for your suggestion. We have now included the following sentences to the manuscript.

The following sentences were added (Line 178-182):

              Through the process of single-cell data downscaling and visualization analysis by using UMAP plots, osteoblast-chondrocyte transitional (OCT) cells and osteoblast-reticular transitional (ORT) cells, which have intermediate properties of typical gene expressions of both, are identified as a type of BMSCs. Among them, Fgfr3 is highly expressed in OCT cluster.

Comment 9:

-Line 224-227, The authors should refer some articles to this description.

Author Response:

              Thank you very much for your suggestion. We have added some references to the manuscript.

The following references were added (Line 244-246):

              Matsushita et al., 2022

Comment 10:

-Line 266, The reference # 57 is inappropriate because this is not an original article but a “Hypothesis” article. Please delete or replace it to proper original articles.

Author Response:

              Thank you very much for your suggestion. We have deleted this reference.

Comment 11:

-Line 272-274, The authors need to change the reference # 59, 60, to some appropriate original articles.

Author Response:

              Thank you very much for your suggestion. We have added some references to the manuscript.

The following references were added (Line 287-290):

              Logan et al., 2004

Comment 12:

-Line 316, The reference # 67 should be replaced because this is a “perspective” article.

Author Response:

              Thank you very much for your suggestion. We have changed references.

The following references were added (Line 338-340):

              Deng et al., 2021

Comment 13:

-In the section of “Characteristics of aging BMSCs”, the authors need to re-consider the reference # 70, 71, 74, and 76 since they are review articles.

Author Response:

              Thank you very much for your suggestion. We have made the necessary revisions to the manuscript. We replaced #70, 74 and deleted #71, 76.

The following references were added (Line 340-342, 350-351):

              Fan et al., 2017; Kobayashi et al., 2000

Comment 14:

-In the section of “Effect of BMSD aging on osteoporosis”, the reference # 79, 81, 82, 87, 88, 89, 91, and 92 should be replaced to proper original articles.

Author Response:

              Thank you very much for your suggestion. We have made the necessary revisions to the manuscript.

The following references were added (the section of Effect of BMSD aging on osteoporosis):

              Fischer et al., 2009; Nishida et al., 1999; Jilka et al., 1996; Chen et al., 2002; Muschler et al., 2001; D'Ippolito et al., 1999

Comment 15:

-Line 391, The authors need to refer an appropriate article to the description of “these delayed premature osteoblasts exhibited activation of cell senescence.”

Author Response:

              Thank you very much for your suggestion. We have added a reference to the manuscript.

The following references were added (Line 417-418):

              Zeng et al., 2022